# Exploring the Microbiome in Gastric Cancer: Assessing Potential Implications and Contextualizing Microorganisms beyond *H. pylori* and Epstein-Barr Virus

**DOI:** 10.3390/cancers15204993

**Published:** 2023-10-15

**Authors:** Wing Sum Shin, Fuda Xie, Bonan Chen, Jun Yu, Kwok Wai Lo, Gary M. K. Tse, Ka Fai To, Wei Kang

**Affiliations:** 1State Key Laboratory of Translational Oncology, Department of Anatomical and Cellular Pathology, Prince of Wales Hospital, The Chinese University of Hong Kong, Hong Kong 999077, China; wsshin@link.cuhk.edu.hk (W.S.S.); xiefuda@link.cuhk.edu.hk (F.X.); chenbonan1994@gmail.com (B.C.); kwlo@cuhk.edu.hk (K.W.L.); garytse@cuhk.edu.hk (G.M.K.T.); kfto@cuhk.edu.hk (K.F.T.); 2State Key Laboratory of Digestive Disease, Institute of Digestive Disease, The Chinese University of Hong Kong, Hong Kong 999077, China; junyu@cuhk.edu.hk; 3CUHK—Shenzhen Research Institute, The Chinese University of Hong Kong, Shenzhen 518000, China; 4Department of Medicine and Therapeutics, The Chinese University of Hong Kong, Hong Kong 999077, China

**Keywords:** gastric cancer, microbiome, *Lactobacillus*, *Streptococcus*, HBV, HCV, *Candida albicans*

## Abstract

**Simple Summary:**

The microbiome emerges as a crucial participant in gastric cancer (GC), a severe cancer burden worldwide. Current understanding has primarily focused on the impact of *H. pylori* and Epstein-Barr virus (EBV) on GC. Still, recent evidence suggests that other influences like microbiota and viral and fungal infections also contribute. This review highlights established factors like *H. pylori* and EBV and other microbes like *Lactobacillus*, *Streptococcus*, hepatitis B virus, hepatitis C virus, and *Candida albicans*. Improved understanding of the gut microbiome may lead to advanced diagnosis and therapies for GC.

**Abstract:**

While previous research has primarily focused on the impact of *H. pylori* and Epstein-Barr virus (EBV), emerging evidence suggests that other microbial influences, including viral and fungal infections, may also contribute to gastric cancer (GC) development. The intricate interactions between these microbes and the host’s immune response provide a more comprehensive understanding of gastric cancer pathogenesis, diagnosis, and treatment. The review highlights the roles of established players such as *H. pylori* and EBV and the potential impacts of gut bacteria, mainly *Lactobacillus*, *Streptococcus*, hepatitis B virus, hepatitis C virus, and fungi such as *Candida albicans*. Advanced sequencing technologies offer unprecedented insights into the complexities of the gastric microbiome, from microbial diversity to potential diagnostic applications. Furthermore, the review highlights the potential for advanced GC diagnosis and therapies through a better understanding of the gut microbiome.

## 1. Introduction

Over the past few decades, gastric cancer (GC) incidence and mortality rates have declined [1]. However, GC remained the fifth most common cancer worldwide and was estimated to cause 769,000 deaths in 2020 [2,3]. GC could be classified into two primary topographical subsites: cardia gastric cancer, which originated near the esophagogastric junction of the stomach, and non-cardia gastric cancer, which emerged from regions farther away from that junction [4].

Significant racial differences have been found in GC incidences. All non-white groups exhibited considerably higher non-cardia GC incidences than non-Hispanic white participants; the incidence was highest in Korean-American men above 50 [5]. Ethnic differences were also present in terms of GC prognosis. Asian patients had significantly longer median survival times than their Caucasian counterparts, especially for early-stage GC, i.e., Stage IA, IB, IA, and IIB [6].

GC is a complex disease that various factors, such as environmental, genetic, and infectious factors, can influence [7]. Previous researchers have extensively studied infective factors such as *Helicobacter pylori* (*H. pylori*) infection and Epstein-Barr virus (EBV) for GC.

While *H. pylori* and EBV have traditionally been the focus of GC research, recent advancements in high-throughput sequencing technology have provided new insights into the gut microbiome in this malignancy.

This literature review aims to give an update on the involvement of the gut microbiota in GC pathogenesis, progression, and treatment response beyond the scope of *H. pylori* and EBV. By examining recent studies on microbial dysbiosis and two hepatitis viruses associated with GC, this review also highlights the potential for improvement in GC treatment methods by harnessing the power of the diverse and complex gut microbiota.

## 2. Bacteria

### 2.1. H. pylori

Three components of the gastric microbiota, namely *H. pylori*, *Lactobacillus*, *and Streptococcus*, have been confirmed to significantly contribute to the progression of gastric cancer through various pathways. These bacteria are not only involved in infections and inflammation that primarily occur in the precancerous stage but also play a role in the immune microenvironment, angiogenesis, and tumor metastasis in cancer. Table 1 provides a comprehensive summary of the detailed effects of gastric microbiome biological processes in gastric tumorigenesis.

GC and *H. pylori* have a well-established and complex relationship. Estimated to have colonized approximately 50% of the world’s population [33], *H. pylori* infection increases the risk of developing both intestinal-type and diffuse-type GC [34]. It is a Gram-negative bacterium that colonizes gastric mucosa [35,36], classified as a class I carcinogen by the World Health Organization, and responsible for approximately 89% of all GC [37]. *H. pylori* infection can lead to the development of GC through two main pathways: indirect effects mediated by inflammatory processes and direct effects on the molecular composition of gastric epithelial cells through the toxic action of virulence factors [38]. Infection with *H. pylori* can lead to chronic inflammation of the gastric mucosa, leading to histological changes, i.e., atrophic gastritis, small intestinal metaplasia, colonic metaplasia, dysplasia, and eventually gastric carcinoma [38,39]. Additionally, *H. pylori*’s virulence factors, such as cytotoxin-associated gene A (CagA) contained with the Cag pathogenicity island (cagPAI) and vacuolating cytotoxin A (VacA), can directly interact with and influence the molecular makeup of gastric epithelial cells, contributing to the initiation of gastric carcinogenesis [38]. Urease, another virulence factor for GC, could facilitate bacterial colonization and modulate host immune responses via mechanisms such as facilitated apoptosis by binding to the class II major histocompatibility complex (MHC) receptors. A high level of urease might induce histopathological alterations within the gastric mucosa and lead to further gastric carcinogenesis [40].

More than 50% of people worldwide have *H. pylori* infection, but the projected lifetime risk of GC is only 1–2 percent [41]. However, due to the large number of infected individuals, many *H. pylori*-infected patients still develop GC. *H. pylori* eradication, primarily using amoxicillin (AMPC) and clarithromycin (CAM) along with proton pump inhibitors (PPI) to suppress gastric acid secretion [42], and *H. pylori* treatment has been recognized as a crucial strategy for achieving multiple benefits. These benefits include reducing proliferation rates [43], and lowering the incidence of GC, especially the intestinal type [44], minimizing the risk of its development [45], especially for patients without precancerous lesions [46], and ultimately reducing mortality from GC [47].

### 2.2. Gastric Microbiota

*H. pylori*, as previously discussed as a risk factor for GC, can modulate the gastric environment [48], also known as the dysbiosis [49]. *H. pylori* neutralizes the gastric acidic environment via urease activity [24,25], reducing gastric acid secretion and promoting chronic inflammation, thereby increasing gastric pH. Gastric acid could be a robust barrier against the invasion of orally ingested microorganisms into the gastrointestinal tract. Elevated gastric pH resulting from *H. pylori* infection might facilitate the overgrowth of non- *H. pylori* microbes within the gastric ecological niche [50]. In particular, the colonization of H. pylori was linked to significantly reduced alpha and beta diversities in the gastric mucosa [51]. Moreover, the virulent genotypes of *H. pylori* are linked to variations in the microbial profile, suggesting that the bacterial virulence of *H. pylori* might influence the composition of the gastric microbiome [52].

There has been insufficient evidence to directly link the non-*H. pylori* bacterial community in the stomach to gastric carcinogenesis [49]. However, with technological advances, more researchers have found significant differences in the gastric microbiota in patients with GC compared to healthy individuals. Microbiota profiles gradually change from gastritis to a pre-neoplastic lesion to GC [53]. Bacterial load is significantly higher in GC patients, indicating that bacterial overgrowth in the stomach is a possible marker for GC [54]. In addition to richness, GC patients have increased diversity in their gastric microbiota [55]. At the phylum level, Firmicutes are significantly higher in GC patients [56]. In terms of genera, some genera of bacteria are enriched in GC patients, while others show a decrease. Table 2 summarizes recent studies conducted in the past six years, focusing on the difference in genera between healthy and GC patients.

As shown in Table 1, several studies support that *Lactobacillus*, a type of lactic acid-producing bacteria, is significantly more abundant in both progressive histological phases of gastric carcinogenesis and patients diagnosed with GC [68]. An average level of *Lactobacillus* exerts a probiotic effect on humans, conferring several benefits to the human stomach. These benefits include excluding pathogens, maintaining the gastric barrier function, anti-inflammatory properties, and potential anti-cancer effects [69]. However, the detailed mechanism of how excess *Lactobacillus* stimulates GC carcinogenesis has remained unclear. Some research has proposed that instead of causing GC development, *Lactobacillus* overgrowth might change the composition of the stomach microbiome through interactions with other bacteria and potentially cause dysbiosis [14].

On the other hand, as summarized in Figure 1, other studies have suggested that an excessive rise in *Lactobacillus* could be detrimental because lactic acid, produced by these bacteria, could act as an energy source for tumor cells and promote tumor proliferation and survival [70]. Consequently, it induces glycolytic enzymes, leading to an enhanced supply of ATP. It might also promote inflammation, tumor angiogenesis, and metastasis, further contributing to the complex dynamics of GC development [71,72].

The oral microbiota affect the gastric microbiota, and this influence occurs in a bidirectional manner [73]. Due to the weakened immunity of GC patients, GC individuals exhibit a more complex oral bacterial profile compared to the average control population. Oral microbiota might even be a future diagnostic tool [74]. A Korean case-control study revealed that the enriched bacterial genera *Streptococcus*, *Abiotrophia*, and *Leuconostoc* and decreased *Prevotella*, *Haemophilus*, and *Neisseria* increased the risk of cancer [75]. This suggested the potential to make use of oral microbiota for the diagnosis of cancer, including GC. In fact, with a cohort of 293 patients in China, a study has already revealed that the salivary microbiota could accurately identify GC among patients with non-malignant gastric diseases such as superficial gastritis and atrophic gastritis [76]. Another Chinese study of 50 participants revealed a highly sensitive scoring system for oral microbiota screening GC [77]. All of these studies concluded that a strong potential exists for oral microbiota as a diagnostic tool.

However, the heterogeneity of oral bacteria in different places must be considered when formulating a GC diagnosis tool based on oral microbiota. GC patients in Singapore and Malaysia had higher abundances of oral bacteria such as *Leptotrichia*, *Fusobacterium*, *Haemophilus*, *Veillonella*, and *Campylobacter* than other regions [78].

Another possible reason behind altered gastric microbiota might be GC treatment methods. Subtotal gastrectomy, a standard surgical treatment for early distal GC, also led to dramatic changes in gastric microbiota. After surgery, the gastric microbiota showed phylum-level shifts, with decreased Proteobacteria and Actinobacteria and increased Firmicutes and Bacteroidetes. Specific genera, such as *Ralstonia* and *Helicobacter*, were abundant in GC patients, while *Streptococcus* and *Prevotella* dominated after surgery. Notably, predicted gene functions also varied, with denitrification and nitrosation genes prevalent before surgery and increased bile salt hydrolase and NO and N_2_O reductases after surgery, indicating metabolic changes in response to the surgical intervention [79].

Recently, contrasting findings about the prevalence of various bacteria with or without GC have been shown. A Chinese study discovered no statistical difference in the abundance and prevalence of *H. pylori* between GC and non-GC individuals [80]. In contrast, two studies from Taiwan and Portugal revealed a decrease in *H. pylori* for GC patients, as shown in Table 2. Moreover, in the United States, a study using shotgun metagenomic sequencing demonstrated a significant decrease in the abundance of *Streptococcus* in the gastric microbiota of individuals with GC compared to those with chronic gastritis. In contrast, several other studies employing 16S rRNA gene sequencing reported a noteworthy increase in *Lactobacillus* and *Streptococcus* in GC relative to chronic gastritis [81]. Contrasting findings have also been shown in terms of microbial diversity. Indeed, some studies indicate a lack of difference between cancer patients and those with functional dyspepsia or chronic gastritis. However, contrasting reports pointed to increased microbial diversity or decreased diversity in cancer patients compared to the mentioned groups [82]. These varying results highlighted the complexity and ongoing exploration of the relationship between gastric microbiota and GC. A very recent study in 2023 by Nikitina et al. reported significant differences in the relative abundance of the gastric tissue microbiome when comparing 16S rRNA gene transcript and 16S rRNA gene levels in all study groups, including control, tumor, and tumor-adjacent samples. This finding suggested that the molecular analysis method might influence the results and findings. The two methods, 16S rRNA gene transcript analysis and 16S rRNA gene analysis, might capture different aspects of the microbial community, leading to variations in the identified taxa and their relative abundance [83]. Despite the controversial impact of gut microbiota on GC development [84], the gastric microbiome might be a future biomarker for GC diagnosis after more extensive research. The potential mechanism of the gastric microbiome contributing to GC development has remained unclear.

## 3. Viruses

### 3.1. EBV

The connection between GC and EBV is well established. EBV, a worldwide oncogenic g-type herpes virus, has been detected in a subset of GC cases (EBVaGC) [85,86], particularly more frequent among tubular adenocarcinomas and medullary carcinomas [87]. It has been suggested that EBV infects both B and epithelial cells [88], causing epithelial cancer, e.g., GC [89]. EBVaGC represents approximately 10% of all GC worldwide [90]. EBV microRNAs are often highly expressed even during the latent infection period. Specifically, the overexpression of EBV-miR-BART5-3p has stimulated the proliferation of GC cells in both laboratory settings and in living organisms [91,92]. BART5-3p targets the tumor suppressor gene TP53 within its 3′-untranslated region, suppressing CDKN1A, BAX, and FAS expression. This accelerates the progression of the cell cycle and inhibited cellular apoptosis. Additionally, BART5-3p contributes to resistance against chemotherapeutic agents and obstructs the increase in p53 induced by ionizing irradiation [92]. This heightened expression of EBV miRNAs generally translates into a threefold increase in the mortality risk associated with GC [93]. While some studies have suggested that EBVaGC patients had a better survival rate than EBV-negative patients [94], another study found that the overall survival of Stage I–III EBVaGC was similar to non-EBVaGC [95]. EBV-related GCs had significant differences in their phenotypic and clinical attributes compared to EBV-negative GC, including the absence of the cyclin-dependent kinase (CDK) inhibitor p16INK4A (encoded by CDKN2A) expression, the presence of unaltered p53, observable driver mutations in the PI3K catalytic subunit-α (PIK3CA) [96], higher PD-L1 [97,98,99] and PD-L2 expression [100], extreme viral and host DNA hypermethylation [101,102,103], and the loss of p16 expression [104]. Transitioning to advanced EBV-associated GC, their histological features differed and had a notably higher rate of aberrant p53 immunostaining patterns than conventional EBV-associated GC [105]. Table 3 summarizes EBV’s mechanism in promoting GC, alongside other viruses.

### 3.2. Hepatitis B Virus (HBV) and Hepatitis C Virus (HCV)

HBV and HCV, which are both transmitted parenterally [106], have been considered potential risk factors for gastric cancer (GC) [107], not solely for hepatocellular carcinoma [108].

#### 3.2.1. Infection of HBV and HCV

From the World Health Organization (WHO), in 2019, approximately 296 million individuals lived with chronic HBV infection worldwide, with an estimated 1.5 million new infections occurring annually [109]. HBV is primarily transmitted through vertical transmission, sexual contact, and blood [110]. However, the main transmission routes vary depending on geographical area. For instance, vertical transmission plays a significant role in Asia, whereas horizontal transmission from child to child is the most frequent in Africa [111]. With the highly effective vaccine against HBV that has been available since 1982, the prevalence of HBV in developed countries has significantly decreased. However, in Sub-Saharan Africa and East Asia, it was estimated that 5–10% of adults still suffer from chronic HBV infection [112]. HBV has been widely known to play a vital role in developing hepatocellular carcinoma [113,114]. Some recent meta-analyses have shown that HBV infection could also significantly increase people’s risk of developing GC [115,116], but the data on such an association were still limited and not robust. A Kailuan cohort study found a nonsignificant association between HBV infection and GC [117]. Another study further suggested that HBV infection alone was not a strong etiology for GC, but its coexistence with *H. pylori* would affect the GC progression significantly [118].

Currently, unlike HBV, no HCV vaccine exists. HCV infection remains a significant global health issue [119]. According to the WHO, approximately 58 million individuals worldwide live with chronic HCV infection, and approximately 1.5 million new infections are reported yearly [120]. HCV is typically transmitted through substantial or recurrent contact with blood, such as transfusion [121]. It was found to be primarily associated with liver-related issues like cirrhosis and could also lead to complications beyond the liver, such as diabetes mellitus [122]. As for its association with GC, current research has not established a direct link between HCV infection and the development of GC. However, it is essential to note that chronic HCV, similar to HBV, has been gradually found to be a possible risk factor for GC [108]. The likelihood of developing GC among patients infected with HCV was observed to be 88% higher compared to individuals without HCV infection (*p* = 0.001) [123].

#### 3.2.2. Molecular Pathogenesis of HBV-Related and HCV-Related GC

The detailed underlying molecular pathogenesis of HBV-related and HCV-related GC remains unknown [124]. However, many previous studies have strongly supported chronic HBV and HCV as an etiology of liver cirrhosis [125,126,127], and liver cirrhosis was recognized as a risk factor of GC [128,129]. This indicates a possible indirect relationship between HBV, HCV, and GC.

Specifically for HBV, as illustrated in Figure 2, upon infection, HBV replicates in the hepatocytes. It does not stay only in hepatocytes but also in extrahepatic tissues [108], including gastric mucosal epithelial cells. HBV DNA has been detected in 12.4% of GC tissues [130]. Based on gastric mucosa epithelial cells also infected with HBV, a study suggests that HBV infection may increase the risk of GC through an analogous mechanism to HBV-related hepatocellular cancer [131]. HBV may cause permanent DNA alteration in the gastric cells by changing the methylation status of cellular DNA, known as a “subthreshold neoplastic state” [132,133]. This, in turn, induces local inflammation in the stomach. Solid evidence supports inflammation as a pivotal factor in etiopathogenesis and ongoing tumor development [134].

In addition, the HBV genome encodes seven proteins, including the X protein (HBx) [135]. HBx is widely established to be a critical factor in the development of numerous types of cancer. HBx is significantly expressed in GC cells but not normal cells [136]. The binding of HBx to various cellular proteins, including hepatitis B virus X-interacting protein (HBXIP) [137], could be attributed to the elevated expression of HBXIP in GC tissues compared to paracancerous tissues [138]. HBXIP regulates cell proliferation and migration through activating NF-κB and IL-8 [139], apoptosis, and cell division and is overexpressed in several cancers [140]. NF-κB promotes survival genes within cancer cells and inflammation-promoting genes in tumor microenvironment components [141]. The expression of IL-8 in GC cells indicates the extent of angiogenesis in the tumor [142]. HBXIP promotes cancer progression by causing a regulatory cascade that influences signaling pathways and the miRNA [143]. HBXIP enhances glucose uptake, particularly in GC, stimulating glycolysis and mitochondrial respiration in GC cells [140]. A study reveals that HBXIP also plays an oncogenic role in GC via METTL3-mediated MYC mRNA m6A modification [138]. Moreover, the anti-HBc antibodies, the most sensitive and reliable seromarker for previous HBV exposure, are expressed highly in GC subjects compared to the control [136,144], supporting a possible association between GC and HBV.

However, immunohistochemical findings have implied that the HBx protein and HBcAg (or HBc protein or HBV core protein) expression are restricted to the cytoplasm instead of being found in the nucleolus. Furthermore, the low detection rate of cccDNA suggests that HBV may exhibit diminished replication activity in tissues outside the liver. These observations provide insights into the potential extrahepatic involvement in the molecular pathogenesis of HBV-related GC. Additional functional analyses and research are necessary to elucidate the detailed mechanisms through which HBV infection and protein expression may influence GC development [136].

The gut microbiota is a significant component of a huge human symbiotic community with trillions of microorganisms [145,146]. Such microbiota is essential in many physiological functions, such as maintaining the intestinal mucosal barrier and immune protection [147]. Notably, the gastric microbiome has been suggested to be a risk factor for GC [71]. The microbial diversity decreases constantly throughout GC progression, and cancer patients and non-cancer individuals may have a significantly different microbial composition [66]. Research employing advanced 16S rRNA gene sequencing techniques revealed a reduced bacterial diversity in individuals suffering from HCV infection. These findings demonstrated a decline in the presence of *Clostridiales* while observing an increase in *Streptococcus* and *Lactobacillus* species [148]. Such microbiome dysbiosis resulting from HCV infection may lead to an increased risk for GC.

#### 3.2.3. Host Immunity of HBV-Related and HCV-Related GC

The host immune system prevents HBV and HCV infection with innate and adaptive immunity [149]. Innate immunity acts as a first line of defense and is typically activated quickly upon exposure to a foreign pathogen [150], including HBV and HCV. Both HBV and HCV are known to activate inflammasomes such as the NLRP3 inflammasome [151,152].

One innate pathway against HBV involves toll-like receptors (TLR3, 7, 8, and 9) in endosomal membranes of specialized cell types, e.g., plasmacytoid dendritic cells, which recognize HBV, as shown in Figure 3A. They then recruit proteins such as MYD88 to activate protein kinases, including IKK-related kinases [153]. A previous GC mouse model also supported that lacking the essential TLR signaling adaptor MYD88 could suppress tumor formation [154]. Nuclear factor kappa B subunit 1 (NF-κB) inhibitor (I-B) is then phosphorylated by the IKK complex, and this causes I-B to be targeted for destruction by the ubiquitin-proteasome pathway, allowing NF-κB to enter the nucleus and activate a wide range of genes involved in immunological and inflammatory responses. IRF3 or IRF7 is phosphorylated by TBK1 and IKK, which causes dimerization and nuclear translocation. Interferons are produced and released to attach to their receptors on HBV-infected cells and nearby non-infected cells by the nuclear IRFs, NF-κB, and other transcription factors, forming an enhanceosome complex. Interferon-stimulated genes (ISGs), which prevent HBV replication, are induced when interferon receptors are engaged by activating the activator of transcription (STAT) signaling pathway and the Janus kinase (JAK)-signal transducer [153]. Researchers have validated the ability of TLRs to inhibit HBV replication through intravenous injection of the panel of ligands specific for TLR2, TLR3, TLR4, TLR5, TLR7, and TLR9 and analyzing HBV DNA via Southern blot analysis [155].

HCV infection has been primarily detected by cytosolic pattern recognition receptors and endosome membrane receptor toll-like receptor 3, the previously mentioned receptor, which promote the synthesis of interferons (IFNs) and inflammatory cytokines through different signaling pathways [156,157,158]. Illustrated in Figure 3B, HCV replicates and accumulated double-stranded RNA (dsRNA) intermediates, which are identified and then activated TLR3. This process subsequently leads to the activation of the toll/interleukin-1 receptor-domain-containing adapter-inducing interferon-β (TRIF). Consequently, the transcription factors IRF3 and NF-κB are activated, promoting the synthesis of interferons (IFNs) and inflammatory cytokines [157].

However, HBV and HCV affect some innate immunity signaling pathways by interfering with various components of the innate immune response. HBV impacts innate immunity signaling by being invisible to pattern recognition receptors [159]. The type 1 interferon response, an effective anti-viral response, typically activates interferon-stimulated genes, which stimulate the effector function of immune cells (e.g., T cells) for eliminating pathogens [160]. A study found that genes encoding critical factors in the type 1 interferon response, such as IFNB, were absent in early HBV-infected organisms [161]. HCV NS3/4A protease effectively cleave and deactivate the mitochondrial anti-viral signaling protein (MAVS) and the toll-interleukin-1 receptor-domain-containing adaptor-inducing interferon-β (TRIF) involved in the detection pathways that respond to HCV pathogen-associated molecular patterns (PAMPs) to stimulate interferons (IFNs), which aid in escaping from the innate immune response [162].

#### 3.2.4. Diagnosis and Treatment of HBV-Related and HCV-Related GC

The prognosis of HBV-related GC patients tends to be poorer than other GC patients. Through Cox proportional hazards regression, a recent study revealed that chronic HBV infection led to poor prognosis in the GC development [163]. HBV-related GC patients were more likely to suffer liver metastasis after GC surgery and poorer disease-free survival and 5-year overall survival rates than GC patients without HBV. Hence, HBV might be an adverse biomarker in predicting GC patient survival rates [163].

Monoclonal antibodies in immune checkpoint blockade therapy have targeted mainly PD-1/PD-L1 proteins [164]. However, patients with HBV-related GC exhibit a reduced expression of programmed cell death ligand 1 (PD-L1), suggesting a decreased possibility of successful immune checkpoint blockade therapy [165]. Immune checkpoint blockade therapy has become a modern standard of care for advanced GC patients [166] and profoundly transformed the landscape of cancer treatment approaches [167,168]. Also, the high expression of HBXIP in HBV-related GC patients has led to higher drug resistance in tumor cells [143], which might bring primary resistance to the immunotherapy [169], decreasing the effectiveness of immune checkpoint inhibitors for HBXIP^+^ GC patients [170].

The prognosis of HCV-related GC has yet to be studied extensively. A population-based retrospective cohort study in Taiwan showed the lowest GC incidence and mortality rates in HCV-uninfected individuals. At the same time, treatment against HCV did not lower the GC risk for patients. This indicated that HCV antiviral treatment might not effectively reduce GC incidences [171].

**Table 3 cancers-15-04993-t003:** Biological processes affected by viruses.

Biological Process	Virus Type	Mechanisms
Infection and Entry Process	EBV	EBV-infected naive B cells, forming proliferating latently infected lymphoblasts expressing latent proteins. In the germinal center, these cells exhibit a confined protein profile, expressing either EBNA1 or remaining in a latent state. A small subset might undergo lytic reactivation under signals at any time, releasing infectious viruses for spread or reinfection [172]
	HBV and HCV	Infected hepatocytes specifically [108,173]; HBV first bound to hepatocytes through low-affinity binding to heparin sulfate proteoglycans and high-affinity binding to the NTCP receptor, then entered through endocytosis. HCV interacts with 14 or more host cell factors for efficient infection [174]
Carcinogenic Factor	EBV	The latent membrane protein 2A (LMP2A) participated in cellular biosynthesis and influenced subsequent genes and cellular behaviors by engaging the AKT and AMPK signaling pathways [175]
	HBV	HBXIP stimulated both cellular growth, migration, and invasion in laboratory settings and living organisms [176]
	HCV	HCV nonstructural protein genes contributed to fibrosis development, which might indirectly promote carcinogenesis, by triggering the synthesis of transforming growth factor beta and activating hepatic stellate cells; HCV core protein might play a role in promoting the development of cancer [177]
Inflammation	EBV	Latent EBV-positive B cells might lead inflammation via upregulated cytokines in EBV-transformed B cells such as TNF-α, TNF-β, and G-CSF [178]. Non-resolving inflammation was conducive to forming a tumor microenvironment for GC tumor initiation and development [134]
	HBV	HBx stimulated toll-like receptor (TLR) and nuclear factor-kappa B (NF-κB) signaling pathways, leading to increased pro-inflammatory cytokine expression; and it triggered NLRP3 inflammasome activation, hastening the release of IL-1β and IL-18 [179]
	HCV	Continued HCV replication within hepatocytes resulted in unregulated inflammation and the production of chemokines [180]. Non-resolving inflammation induced by different viruses was conducive to forming a tumor microenvironment for GC tumor initiation and development [134]
DNA Damage	EBV	Increased DNA hypermethylation of PD-L1/2 [181]
	HBV	Within HBV-infected cell nuclei, the transformation of genomic viral DNA into transcriptionally active episomal DNA (cccDNA) or transcription of viral mRNAs from cccDNA relied on cellular proteins’ enzymatic activities. HBV DNA integration into host chromosomal DNA and the accumulation of mutations in host DNA potentially triggered carcinogenesis [182]
	HCV	The chronic inflammation caused by hepatitis C virus (HCV) might trigger oxidative stress, potentially hindering the repair of DNA damage. This could increase the vulnerability of cells to spontaneous or mutagen-induced changes [183]
Immune Evasion and Immune System	EBV	Evaded immune responses by utilizing lytic gene products, such as BGLF5, which diminished the levels of innate immune EBV-sensing TLR2 and the lipid antigen-presenting CD1d molecule [184]
	HBV	Did not induce significant innate immune activation via pro-inflammatory cytokines and interferons (IFN) [185]
	HCV	To evade the initiation of interferon regulatory factor 3 (IRF3) and subsequent interferon (IFN) production triggered by RIG-I/MDA5, the HCV protease NS3/4a has developed a dual role: processing the HCV polyprotein and cleaving the vital adapter protein MAVS, essential for relaying RIG-I/MDA5-initiated signals, as well as TRIF, which mediates downstream signaling from toll-like receptor (TLR)3 that recognizes double-stranded RNA within endosomes. Consequently, this obstructs the expression of type I and III IFN [185]
Angiogenesis	EBV	The components produced by EBV could encourage the development of tumor angiogenesis through the PI3K/AKT signaling pathway [186]
	HBV	HBx could induce an angiogenic response by directly stimulating angiogenesis alone; and it stimulated angiogenesis through both the transcriptional activation and stabilization of HIF-1α [187]
	HCV	Caused higher serum concentrations of the angiogenic proteins placenta growth factor (PlGF) and Ang-2 [187]
Cellular Proliferation	EBV	Expressed viral oncogenes that facilitate cell growth and hinder the apoptotic response, leading to uncontrolled cell proliferation [188]
	HBV	Induced cellular proliferation in part via HBx-induced miRNA-21 expression [189,190]
	HCV	Inhibited cell proliferation via overexpression of HCV E2 [191]
Metastasis	EBV	The elevated expression of Indian Hedgehog gene increased metastatic potential via angiogenesis and Snail protein expression [192]
	HBV	Hepatitis B surface antigen (HBsAg) was associated with an elevated risk of distant metastasis in cancer patients [193]
	HCV	Proteins encoded by HCV could directly trigger cellular functions that promote metastasis [194]

### 3.3. Human Papillomavirus (HPV)

HPV is a sexually transmitted virus with a well-established role in cervical cancer [195]. While specific high-risk subtypes, such as HPV type 16 [196], have been extensively studied for their role in cervical cancer, their potential involvement in other cancers, including GC, is gaining attention.

#### 3.3.1. HPV Infection

Repetitive and persistent HPV infection was assumed to lead to precursor lesions like dysplasia or adenocarcinoma in situ, eventually resulting in malignancy [197]. Under such an assumption, it was suggested that HPV may enter the anus and colorectum through transmission from anogenital sites [197]. Alternatively, oral entry is possible, leading to downward infection from the mouth to the esophagus and ultimately to the stomach [197].

#### 3.3.2. HPV Gastric Oncogenesis

HPV has been believed to contribute to neoplastic changes in the gastric mucosa, potentially resulting in the advancement of GC [198]. HPV increases gastrointestinal cancer risk, particularly of squamous cell origin [199]. HPV oncoproteins, specifically E6 and E7, disrupt cellular pathways regarding cell cycle regulation, apoptosis, and cell polarity control networks [200]. E6 affects cell cycle regulation and stimulates excessive cellular proliferation by exerting its inhibitory effect on p53 through different pathways. One mechanism involves the disruption of E6AP, a ubiquitin ligase that facilitates proteasome-dependent degradation. E6 forms a complex with E6AP and p53 in this process to promote p53 degradation [201]. Such an HPV oncogenesis mechanism might lead to GC tumor initiation and progression.

#### 3.3.3. Clinical Significance of HPV in GC

The clinical significance of HPV in gastric cancer remains an active area of investigation. Studies have shown conflicting results regarding HPV positivity in gastric cancer patients. According to a meta-analysis, the overall prevalence of HPV in gastric cancer (GC) was 23.6%. Interestingly, there were significant variations in HPV status between Chinese and non-Chinese studies. Notably, HPV positivity was 1.43-times higher in Chinese GC patients than in non-Chinese patients, indicating a potential association between HPV infection and GC development in the Chinese population [202]. However, another study did not detect HPV DNA in the gastric mucosa tissue of GC patients in China [203]. Further large-scale studies and rigorous analyses are required to determine any distinct clinicopathological features and response patterns to treatment associated with HPV-positive gastric tumors to improve clinical practices for better patient outcomes.

### 3.4. Other Types of Viruses

Apart from HBV, HCV, HPV, and the well-studied EBV, many other oncoviruses are still suspected to be a risk for GC. A meta-analysis in 2020 revealed that patients with GC exhibited a notably increased viral load of human cytomegalovirus (HCMV) [204], suggesting a possibility that it might also play a role in GC progression. The role of HCMV in oncogenesis has remained a subject of debate since most infected individuals do not develop cancer, and the virus alone is generally not potent enough to initiate tumor formation [205].

## 4. Fungi

Fungi have played a notable role in gastric carcinogenesis. These eukaryotic microorganisms have been widely distributed globally, with two hundred orders and a dozen phyla. Particularly, a few hundred fungal species that belonged to a few lineages have caused most related infections and deaths [206].

Intriguingly, despite the acidic nature of the stomach, many species of fungi have been identified in the healthy human gut. However, only a limited number of fungi can thrive and establish colonization within the gut [207]. *Candida* and *Phialemonium* were the main fungi among them [208]. Among fungal species in a study using endoscopic and mycological examinations, *Candida albicans* was isolated most frequently in normal subjects at 48%, and *Candida tropicalis* came second at 29.3% [209]. Many recent studies have shown that fungi infection, especially *Candida albicans*, a member of the microbiota that could cause infection under certain circumstances, increases the host risk of GC [210,211,212].

Most pathogenic fungi are promptly identified and eliminated by innate immune cells with diverse pattern-recognition receptors. Typically, these fungi remain harmless unless there is a disturbance in the immune equilibrium [213]. Factors for such disturbances include deficiencies and polymorphisms in pattern recognition receptor systems [214], systemic immunosuppression due to conditions like acquired immune deficiency syndrome (AIDS) [215], leukemia [216], diabetes mellitus [217], solid organ transplantation [218], and long-term mechanical ventilation [219]. Significant gastric fungal imbalance was also found in GC. *Candida* and *Alternaria* were elevated in GC patients at the genus level, while *Saitozyma* and *Thermomyces* (*p* = 0.009) ere decreased. In particular, *Candida albicans* were significantly increased in GC [211].

It is crucial to acknowledge the significant morbidity and mortality risks associated with opportunistic *Candida* infections in GC patients [220]. While fungal microorganism infections in the context of GC are relatively rare, when they do occur, they can have severe consequences. Therefore, it is imperative to recognize the potential impact of fungal microorganisms, such as *Candida*, and emphasize the importance of studying their role in GC. By understanding their contribution to disease progression, we can advance our knowledge and potentially identify new therapeutic approaches to improve patient outcomes.

## 5. Conclusion and Future Perspectives

The rapid advancements in understanding the microbiome’s role in GC have opened up new avenues of research and potential therapeutic interventions. While most existing research has been directed towards established factors such as *H. pylori* and EBV, emerging evidence suggested that a broader spectrum of microbial factors, including viral and fungal infections, could contribute to gastric carcinogenesis. This literature review delved into the intricate interplay between the gut microbiome and GC, focusing on familiar and lesser-explored microbes.

*H. pylori* has remained a significant player in gastric carcinogenesis. This Gram-negative bacterium was intricately linked with chronic inflammation and molecular alterations that drive oncogenic processes. While extensive research has unveiled its association with DNA damage, oncogene activation, and tumor suppressor inhibition, recent high-throughput sequencing techniques allowed us to delve deeper into its influence on the gastric ecosystem. The modulation of the gastric environment through *H. pylori*’s activities could pave the way for the overgrowth of non*-H. pylori* microbes within the stomach, altering the microbial balance and potentially influencing cancer development.

Furthermore, EBV has long been associated with GC, particularly the unique subset known as EBVaGC. This oncovirus exerts its influence through complex molecular mechanisms, including miRNA-initiated methylation pathways. Recent immunotherapy advances have provided avenues for targeted treatment in this specific subtype. However, the relationship between EBV and GC is far from one-dimensional, with differing clinical and phenotypic attributes highlighting the need for nuanced investigations.

Including HBV and HCV in the discussion further broadens our understanding of viral influences on GC. Chronic infections with HBV and HCV have traditionally been associated with hepatocellular carcinoma, but their potential role in gastric carcinogenesis has also garnered attention. While the precise molecular mechanisms linking these viruses to GC remain unclear, the interaction between these viruses and the host immune response could be crucial in shaping disease outcomes.

Interestingly, the less-explored territory of fungal infections in GC is also gaining traction. *Candida albicans*, a common fungus in the human digestive tract, has been implicated in gastric carcinogenesis. Research has suggested that an overgrowth of *Candida albicans* in the stomach could potentially impact the development of GC. Various factors could influence this fungal imbalance, including systemic immunosuppression and other underlying health conditions.

Moving forward, several intriguing areas could be explored. The emergence of high-throughput sequencing technologies offers unprecedented opportunities to comprehensively profile the gastric microbiome in health and disease. Understanding the dynamics of microbial diversity, abundance, and interplay could shed light on the intricate relationship between microbes and GC. The potential of the gut microbiome as a diagnostic tool is particularly promising, with studies hinting at its role in distinguishing healthy individuals from those with precancerous lesions or malignancies.

Moreover, therapeutic avenues could emerge as we uncover the molecular intricacies of viral and fungal interactions. Expanding the scope of immunotherapies to encompass a broader spectrum of infections and focusing on modulating the gut microbiome could hold promise for more effective treatment strategies. Additionally, unraveling the role of microbes in influencing treatment responses, such as immune checkpoint blockade therapies, could pave the way for personalized approaches to GC management.

In conclusion, the landscape of microbial influences on GC is evolving rapidly. Beyond the established role of *H. pylori* and EBV, viruses such as HBV and HCV and fungal species like *Candida albicans* are emerging as potential contributors to gastric carcinogenesis. However, these viruses’ studies are still in their infancy, and the detailed molecular mechanism remains unclear. More deep mechanism studies on the interaction between these oncoviruses and GC cells are required to shed light on the treatment methods targeting the microbiome.

## Figures and Tables

**Figure 1 cancers-15-04993-f001:**
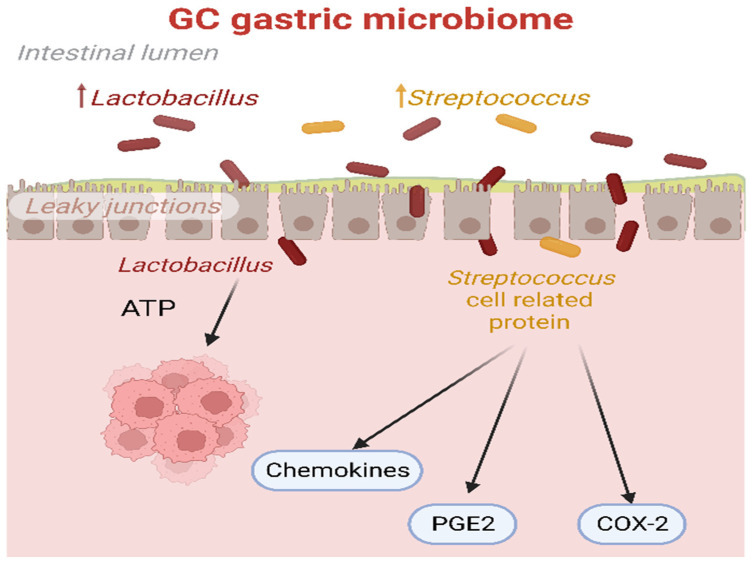
The biological processes triggered by *Lactobacillus* and *Streptococcus* in the GC microbiome. Lactic acid produced by *Lactobacillus* could serve as a potential energy source for tumor cells. Additionally, proteins associated with *Streptococcus* cells could induce the release of chemokines, PGE2, and over-expression of COX-2, thereby promoting the angiogenesis process in cancer. COX-2: prostaglandin-endoperoxide synthase 2; PGE2: prostaglandin E2.

**Figure 2 cancers-15-04993-f002:**
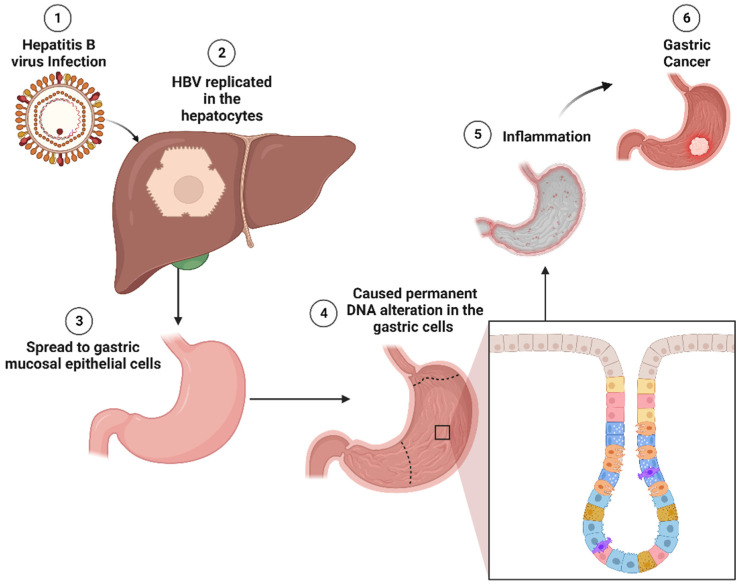
Molecular pathogenesis of HBV-induced GC. HBV replicates primarily in hepatocytes and can spread to gastric mucosal epithelial cells. In gastric cells, HBV has the potential to induce permanent DNA alterations by modifying the methylation status of cellular DNA. These alterations may trigger inflammation in the gastric tissue, ultimately contributing to gastric tumorigenesis. HBV: hepatitis B virus; DNA: deoxyribonucleic acid.

**Figure 3 cancers-15-04993-f003:**
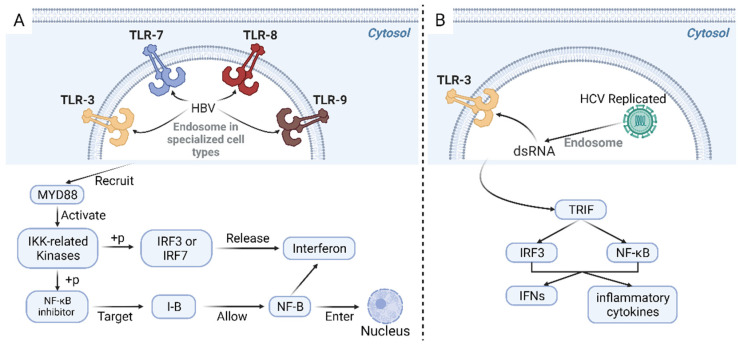
Innate pathway in response to HBV infection (**A**). TLR3, 7, 8, and 9 are receptors located on endosomal membranes of specific cell types, responsible for the recognition of the presence of HBV. These receptors recruit proteins such as MYD88, activating protein kinases, including the IKK-related kinases. The IKK complex phosphorylates the NF-κB inhibitor I-B, targeting it for degradation through the ubiquitin-proteasome pathway. This degradation allows NF-κB to enter the nucleus and activate various genes involved in immunological and inflammatory responses. Additionally, phosphorylation of IRF3 or IRF7 by TBK1 and IKK leads to their dimerization and translocation into the nucleus. This event triggers the production and release of interferons, which bind to their receptors on both HBV-infected cells and nearby non-infected cells, forming an enhanceosome complex involving the nuclear IRFs, NF-κB, and other transcription factors. Innate pathway in response to HCV infection (**B**). The virus replicates and accumulates dsRNA intermediates during HCV infection. These dsRNA intermediates are recognized by TLR3 and subsequently initiate a signaling cascade. This cascade activates the TRIF. Consequently, the transcription factors IRF3 and NF-κB are activated, promoting the synthesis of IFNs and inflammatory cytokines. TLR: toll-like receptors; HBV: hepatitis B virus; MYD88: myeloid differentiation primary response 88; IKK: IκB kinase; NF-κB: nuclear factor kappa-light-chain-enhancer of activated B cells; IRF: interferon regulatory factor; TBK1: TANK-binding kinase 1; dsRNA: double-stranded RNA viruses; TRIF: TIR-domain-containing adapter-inducing interferon-β; IFNs: interferons.

**Table 1 cancers-15-04993-t001:** Biological processes affected by gastric microbiome.

Biological Process	*H. pylori*	*Lactobacillu* *s*	*Streptococcus*
Infection and Entry Process	At the beginning, urease neutralized the acidic stomach; made use of flagella to reach gastric epithelial cells; interaction between bacterial adhesins and host cell receptors [8]	Obtained through oral administration [9]; intrinsically resistant to acid (generally below pH 3.0) [10]	Naturally existed in the digestive tract [11]
Carcinogenic Factor	cagA: entered gastric epithelial cells through bacterial type IV secretion; disrupted various host cell signaling routes by serving as an external scaffold or hub protein [12]. vacA: inhibited parietal cells ability to produce acids, leading to hypochlorhydria; supported the growth of nitrate-producing bacteria, which typically did not thrive in the naturally acidic conditions of the stomach [13]	Did not directly cause carcinogenesis but contributes to it indirectly by influencing changes in the gastric microbial community [14]	Unclear carcinogenic mechanism; increased e.g., interleukin-8 (IL-8), cyclooxygenase 2 (COX2) might be carcinogenic factors towards GC [15]
Inflammation	Enhanced the expression of numerous pro-inflammatory cytokines, including interleukin (IL)-1, IL-6, IL-8, TNF-α, NF-κB, promoting gastric inflammation [16]	Normal level of *Lactobacillus* presented anti-inflammatory effects on HT-29 cells by modulating JAK/STAT and NF-κB signaling pathways [17]	To our best knowledge, there has been limited understanding of the precise mechanism by which *Streptococcus* causes inflammation in the stomach. However, several case studies have associated *Streptococcus* with acute gastritis [18]
DNA Damage	Upon oxidative DNA damage in gastric cancer cells, activation of the DNA damage response pathway occurred. Particularly, Chk1 and Chk2 phosphorylation signified the activation of Chk1 and Chk2, which could stop the cell cycle, potentially leading to mitotic exit and genomic instability [19]	Aided in the restoration of DNA damage caused by reactive oxygen species generated by bile [20]	Came across reactive oxygen species generated by the host’s innate immune response against persisters, culminating to DNA damage [21]
Immune Evasion and Immune System	Evaded pattern recognition receptor detection by evading recognition by toll-like receptors (TLRs) and inhibiting signaling mediated by c-type lectin (DC-SIGN) [22]	Enhanced immune system through strengthening the cytotoxic impact of natural killer (NK) cells and impacted the production of various essential pro-inflammatory cytokines, such as IL-1β, IL-4, IL-5, IL-6, IL-8, and IL-13 [23]	The breakdown of inflammatory hyaluronan fragments produced by the host into disaccharides enabled Group B *Streptococcus* to avoid being detected by the immune system [24]
Angiogenesis	H. pylori infection upregulated the expression of angiogenic factors produced by GC cells including VEGF, interleukin-8, and platelet-derived endothelial growth factor. Specifically, the standard H. pylori strain NCTC11637 significantly raised VEGF expression in gastric epithelial cells like SGC7901 and MKN45, which was achieved by enhancing the expression of COX-2 [25]	Produced high amounts of lactic acids and other metabolites that inhibit angiogenesis of tumor growth via downregulating COX2 expression [23]	As illustrated in Figure 1, *Streptococcus* cell-related proteins induced chemokine release, prostaglandin E2 (PGE2), and COX-2 over-expression, which promoted cancer angiogenesis [26]
Cellular Proliferation	Stimulated the production of MMP9 in gastric cancer cells via the semaphorin 5A-mediated ERK signaling pathway, and also stimulated the proliferation, growth, migration, and invasiveness of gastric cancer cells through its effects on semaphorin 5A [27]	Produced lactic acid bacteria that could exert cytotoxic effects by impeding the proliferation of cancer cells [28]	Streptococcus-infected cells mediated by extracellular-matrix-induced cell proliferation [29]
Metastasis	Increased the expression of HPA, which might be linked to MAPK activation, to encourage the invasion and metastasis of GC [30]	They adjusted the microenvironment to hinder cancer metastasis [31] when not in excessive growth	Compromising vascular integrity by reducing adhesion molecules in endothelial cells, aiding the transendothelial migration of tumor cells and promoting metastasis [32]

**Table 2 cancers-15-04993-t002:** Recent studies (2018–2023) of the gastric microbiota in GC and the non-GC stomach.

Year	Region/Country	Method	Genera Increased (↑) and/or Decreased (↓) in GC versus Non-Cancer	References
2018	Taiwan, China	16S ribosomal DNA analysis	*H. pylori* ↓ *Clostridium*, *Fusobacterium*, and *Lactobacillus* ↑	[57]
2018	China	16S rRNA gene analysis	*Peptostreptococcus*, *Streptococcus*, *Parvimonas*, *Slackia*, *Dialister* ↑	[58]
2018	Portugal	16S rRNA gene profiling	*Helicobacter* ↓; *Citrobacter*, *Clostridium*, *Lactobacillus*, *Achromobacter* and *Rhodococcus* ↑	[59]
2019	Korea	Metagenomic 16S rRNA gene sequencing	*Helicobacteraceae*, *Propionibacteriaceae*, and *Prevotellaceae* ↑	[60]
2019	China	16S rRNA gene sequencing	Higher species richness; Lower butyrate-producing bacteria; Other symbiotic bacteria, especially *Lactobacillus*, *Escherichia*, and *Klebsiella. Lactobacillus* and *Lachnospira* ↑	[61]
2019	China	16S rRNA Gene Amplification Sequence Processing	*Acinetobacter*, *Bacteroides*, *Haemophilus parainfluenzae* ↑	[62]
2019	China	16S rRNA gene sequencing	*Escherichia/Shigella*, *Veillonella*, *and Clostridium XVIII* ↑; *Bacteroides* ↓; Both groups with very low abundance of *Helicobacter*	[63]
2020	Mongolia	16S rRNA gene amplicon sequencing	*Lactobacillus* ↑ *Enterococcus* ↑	[64]
2021	China	16S rRNA gene sequencing	Higher microbial diversity; 27 genera e.g., *Leptotrichia*, *Fusobacterium*, *Prevotella*, *Porphyromonas*, *Capnocytophaga Lactococcus*, *Streptococcus* ↑	[65]
2022	Korea	16S rRNA gene profiling	*Verrucomicrobia*, *Deferribacteres*, and *Lachnospiraceae* NK4A136 group ↓	[66]
2023	/	Integration of RNA-Seq data	*Gemella*, *Pseudomonas*, *Acidovorax* ↑	[67]

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
