# Peer review of "Exploring the Microbiome in Gastric Cancer: Assessing Potential Implications and Contextualizing Microorganisms beyond H. pylori and Epstein-Barr Virus"

_cancers, 2023, doi:10.3390/cancers15204993_

Round 1

Reviewer 1 Report

The author provided a comprehensive review of the cause of gastric cancer, based on the H Pylori, EB virus, HCV Candida, and so on.  The mechanism is novel, especially for HCV and Fungi. 

Major

1. Although the author described the cause of GC based on the H. pylori and EB Virus, there is no illustrated figure of the mechanism. As described in table 2, infection, Carcinogenic Factors, and Inflammation are the factors. Please demonstrate the mechanism in the figure, not just the table. 

2. No epidemiological section exists in this review. Gastric cancer has racial differences.  Please summarise or describe the point, especially the difference between Asians and Caucasians.

Reviewer 2 Report

The manuscript titled "An update of the microbiome in gastric cancer: new driving force apart from H. pylori and Epstein-Barr virus" is designed as a review that should highlight the role of established players and new microorganisms with potential impact on oncogenesis in gastric cancer (GC). While the subject is interesting enough, numerous issues must be considered here. 

First, the title is misleading since these new potential players are in no way a driving force since there is very little evidence that they play any actual role in oncogenesis. Many of them were only found as members of the microbiome in some parts of the digestive system or cancer itself, but it is hard to tell if they are merely present as a consequence of the new tumour microenvironment. Thus, the authors should cautiously address their presence or absence in this milieu. Also, the link between some viruses and fungi with GC development is not substantiated with reliable evidence but is indirect, which should be addressed adequately in the text. Some of the explanations of these links are unclear throughout the text.

There are numerous repetitions in the text. There is no need to explain in detail the role of H.pylori and EBV in GC in the Introduction since there are separate chapters dedicated to both of these microorganisms. In Chapter 2.1, the fact that H.pylori infects 50% of the World's population is mentioned twice with the same reference. In more than one place, the new sentence brings a fact already established in the previous sentences. The text is not written in a way that follows the expected mind flow.

The oral microbiota may deserve a subparagraph of their own.

The reference to Table 2 should begin in Chapter 2.1 when the role of H.pylori is being discussed. Table 2 is challenging to follow in this form, broken between pages.

All Latin names of microorganisms should be in italics. 

Human papillomavirus has a more extensively investigated role in the oncogenesis of GC than HBV and HCV, yet it is only briefly mentioned with other types of viruses.

The sentence in Chapter 3.2 is incomprehensible.

The statistical data about HBV infection are either incorrect or outdated. Two billion people worldwide are not infected with HBV but only have evidence of past infection (in the form of antibodies). One of the latest estimates is that there are 296 million chronically infected individuals, but for the latest statistical data, the authors should refer to WHO data. Also, the main transmission routes vary depending on geographical area, which should be considered. The same should be checked for HCV statistics.

In Chapter 3.2.2, it is stated that cirrhosis is recognised as a cause of GC. While cirrhosis can be an environment where cancer can develop, it can not be named the cause of CG.

The HBV genome encodes seven, not four proteins, but has four open reading frames.

Unclear sentence: "HBx binds many cellular proteins, such as hepatitis B virus X protein (HBXIP) [128], which may be explained by the higher expression of HBXIP in GC tissues than in paracancerous tissues."

There is a difference between anti-HBc antibodies and anti-HBc proteins. Antibodies are seromarkers of past exposure to HBV, and anti-HBc protein is actually an HBcAg whose expression can be measured in tumour tissue.

In the first paragraph of Chapter 4, many studies that show an increased risk of GC associated with some fungi are mentioned, and there are no references. Again, the order of sentences is not clear – first, the Candida spp is mentioned, and then some other fungi and Candida again.

These conclusions about the role of fungi are unclear: "Although instances of fungal microorganism infection in the context of GC remain relatively rare, it's crucial to address opportunistic Candida infections to mitigate their significant morbidity and mortality risks related to GC. Such recognition underscores the potential impact of fungal microorganisms in GC."

Some of the references were not complete (example: reference 114).

The language needs the attention of a native-speaker scientist.

Reviewer 3 Report

The manuscript reviewed the other risks of GC besides of H. pylori and EBV. Currently, more and more research is focusing on these. It is important for researchers in this field. The review is comprehensive and includes evidence in different directions. Here are some comments:

1. Since there is no line number, I revised and commented on the pdf file. 

2. All of the bacterial names should be italic

3.  The whole manuscript is written in the present tense, which should be past tense.

4. for the other part, please check the pdf file.

The language is OK, but there are still some typos and some sentences need to be rephrased to reduce the misunderstanding.

Round 2

Reviewer 1 Report

The author made appropriate corrections. The figures are beautiful and suitable.  I have no more comments on this manuscript. 

Author Response

Thank you for your comments!

Reviewer 2 Report

The manuscript “ Exploring the Microbiome in Gastric Cancer: Assessing Potential Implications and Contextualizing Microorganisms Beyond  H. pylori and Epstein-Barr Virus” has been improved upon revision. There are still minor points to be considered:

Line 227 and line 244 – There should be references to WHO sites where the statistical data were found.

Line 293 - Anti-HBc antibodies are seromarker of previous HBV exposure but can be found in serum, not tissues.

Line 296 – HBX protein should be written as HBx to avoid confusion and mixing with HBXIP, which is a cellular protein.

Line 297 – Even if it was commercially called “anti-HBc protein” to mark that it is a protein which reacts with anti-HBc antibodies, there is no “anti-HBc protein” in HBV infection. It can only be HBcAg or HBc protein or HBV core protein.

Line 411 - HPV is not a sexually transmitted illness but a sexually transmitted virus.

Line 464 – Candida is a microbiota member but can cause infection under certain circumstances. This should be first explained here.

Line 474 - Candida and Alternaria should be written in italics.

Line 478 – The conclusion is still not clear. How can Candida have significant morbidity and mortality risk with GC if it is rare? Also, the last sentence is still confusing: “Recognizing the potential impact of fungal microorganisms underscores the importance of studying their role in GC.”

 Moderate editing of English language is still required.
